# Extracellular Expression of Feruloyl Esterase and Xylanase in *Escherichia coli* for Ferulic Acid Production from Agricultural Residues

**DOI:** 10.3390/microorganisms11081869

**Published:** 2023-07-25

**Authors:** Jiaxin Lan, Shujie Ji, Chuanjia Yang, Guolin Cai, Jian Lu, Xiaomin Li

**Affiliations:** 1School of Biotechnology, Jiangnan University, 1800 Lihu Road, Wuxi 214122, China; 6200202022@stu.jiangnan.edu.cn (J.L.); jishujie98@126.com (S.J.); 6220202023@stu.jiangnan.edu.cn (C.Y.); glcai@jiangnan.edu.cn (G.C.); jlu@jiangnan.edu.cn (J.L.); 2National Engineering Research Center of Cereal Fermentation and Food Biomanufacturing, Jiangnan University, 1800 Lihu Road, Wuxi 214122, China

**Keywords:** ferulic acid (FA), feruloyl esterase, signal peptides, de-starch wheat bran, release rate

## Abstract

There is still a large amount of ferulic acid (FA), an outstanding antioxidant, present in agricultural residues. Enzymatic hydrolysis has been regarded as the most effective way to release FA. This present study therefore selected feruloyl esterase (FAE) and xylanase (XYN) from the metagenomes of a cow rumen and a camel rumen, respectively, for their recombinant expression in *Escherichia coli* BL21 and further application in releasing FA. After screening the candidate signal peptides, the optimal one for each enzyme, which were selected as SP1 and SP4, respectively, was integrated into the vectors pET22b(+) and pETDuet-1. Among the generated *E. coli* strains SP1-F, SP4-X, and SP1-F-SP4-X that could express extracellular enzymes either separately or simultaneously, the latter one performed the best in relation to degrading the biomass and releasing FA. Under the optimized culture and induction conditions, the strain SP1-F-SP4-X released 90% of FA from 10% of de-starched wheat bran and produced 314.1 mg/L FA, which was deemed to be the highest obtained value to the best of our knowledge. This result could pave a way for the re-utilization of agricultural residues and enhancing their add-value.

## 1. Introduction

Ferulic acid (FA) is widely distributed in grains, fruits, vegetables, and agroindustry-derived byproducts, accounting for the highest dry weight of cells of up to 3% [1]. The presence of hydroxyl groups and the multi-conjugation of its molecular structure endows a FA-distinct antioxidant activity, such as stabilizing free radicals generated by ultraviolet light, peroxidase, and other metabolic activities [2], and protecting the DNA and cell membranes from damage. At the same time, FA also exhibits antibacterial, anti-inflammatory, anti-aging, anti-diabetic, and neuroprotective effects [3]. As a result, FA has been used in healthcare products and across the cosmetics industry, and has become a potential substance for the treatment of many oxidation-related diseases, including Alzheimer’s disease, cancer, cardiovascular disease, diabetes, and skin diseases [4]. In addition, FA is also the important precursor for aromatic compound synthesis, such as vanillin and 4-vinylguacol [5].

In the plant cell wall, FA mainly exists in its bound form and acts as a cross linker between arabinoxylan and a lignose. It could be obtained using the chemical extraction approach and enzyme hydrolysis strategy; the latter one is comparatively more environmentally friendly and specific [6]. Feruloyl esterase (FAE) could hydrolyze the ester bonds between FA and polysaccharides. Therefore, FAE from various microbiological resources have been applied to achieve the release of FA from agricultural residues, such as maize straw, de-starched wheat bran, and bagasse [7,8,9]. It was found that by synergistic reaction with xylanase (XYN), FAE is able to gain access to the ferulic acyl oligosaccharides more easily, achieving a higher release rate and yields [10]. Since the lignocellulose is not able to go through the cell wall easily and combine with the endogenous enzymes for deep hydrolysis, the extracellular expression of FAE and XYN would be more efficient.

The ruminal metagenomes of the cow and camel have been considered as treasure troves of glycoside hydrolases. The *fae* (Genbank accession no. HQ338149.1) obtained from the China Holstein cow ruminal metagenome has been verified to produce FAE with a high thermal and pH stability, as well as a broad substrate specificity, making it a good candidate to produce FA by degrading biomass [11]. The *xylCM5* gene (Genbank accession no. KX644152.1) obtained from the camel ruminal metagenome was found to encode a β-1,4-endoxylanase XylC with a xylanase domain Xyn, a polysaccharide-binding domain CBM, and a carbohydrate esterase domain CE. Both the wild XylC enzyme and the truncated Xyn-CMB form exhibited a great activity towards the oat xylan within a broad pH range, and the truncated Xyn-CMB form showed a greater thermal stability [12].

In the present study, the full-length *fae* and the partial sequence *xyn-cbm* of the *xylCM5 gene* were fused with their optimal signal peptides (SPs) for their extracellular expression in *Escherichia coli* [13,14,15,16]. The generated recombinant *E. coli* strains were used as the cell factory for producing FA from wheat bran, one of the current agricultural residues that are still used as the low-value feeding stuff. This work could enhance the value of the use of wheat bran and pave the way for a greener and more effective production of FA.

## 2. Materials and Methods

### 2.1. Strains, Plasmids, Medium, and Chemicals

All the *E. coli* strains and plasmids used in this study are listed in Table 1. *E. coli* JM109 and BL21 (DE3) was used for DNA manipulation and protein expression, respectively. The *E. coli* strains were routinely cultured with Luria–Bertani (LB) medium at 37 °C, with ampicillin supplemented at a final concentration of 100 μg/mL when required. The Ezup bacterial genomic DNA extraction kit, Rapid Taq Master Mix, restriction enzymes, and T4 DNA ligase were purchased from TaKaRa Biotech Co., Ltd. (Tokyo, Japan), Vazyme Biotech Co., Ltd. (Nanjing, China), and Sango Biotech Co., Ltd. (Shanghai, China). FA, arabinoxylan, and methyl ferulate (MFA) standards were purchased from Sigma-Aldrich (Saint Louis, MO, USA). The wheat bran sample was provided by Jiangsu Agribusiness group Corp Ltd. All other chemicals were purchased from Sinapharm (Beijing, China).

### 2.2. Cloning and Expression of FAE and XYN

In order to clone the two enzymes of interest, the genes *fae* and *xyn* (*xyn-cbm* of the *xyl*CM5 gene) were synthesized according to the *E. coli* preference of codon usage and ligated with pET22b(+) to fuse the signal peptide pelB with these two enzymes. The SP1~SP5 were then substituted with pelB through fusion PCR, using pET22b-fae and pET22b-xyn as their templates, respectively (Table 1). The PCR products were digested with *Nco* I and *Eco*R I, or *Nde* I and *Xho* I, and then ligated with pETDuet-1, obtaining the SPs-*fae* and SPs-*xyn* expression plasmids, respectively. In order to integrate OsmY, a linker coding the flexible peptide GSGSGS was used. The pETDuet-1-derivative plasmids (Table 1) were further transformed into *E. coli* BL21 (DE3), verified using colony PCR, and stored in 20% glycerol at −80 °C. All the primers used in this study are listed in Appendix A.

To select the optimal SP, the recombinant *E. coli* strains were inoculated, cultivated, and induced with 0.2 mM isopropyl-β-D-thiogalactopyranoside (IPTG) at 30 °C, 150 rpm, when the OD_600_ reached 0.6. The samples were then taken and the subcellular fraction’s enzymatic activity of each recombinant strain were monitored after 12 h of cultivation. BL21, containing the blank pETDuet-1 vector, was used as the negative control. Subcellular fractionation of recombinant FAE and XYN was achieved using the osmotic shock method to isolate the cytoplasmic, periplasmic, and cytoplasmic cell fractions [17].

The *fae* and *xyn* genes, with their optimal SPs, were integrated into the same pETDuet-1 vector, which was then transformed into the BL21 strain.

### 2.3. Activity Determination of FAE and XYN

To determine the FAE activity, 100 μL of purified FAE was mixed with 900 μL 1 mM methyl ferulate in 0.05 M citric-Na_2_HPO_4_ (pH 6.0) to react at 40 °C for 10 min [18]. The reaction was then terminated with 400 μL acetic acid. After extraction with methanol and filtration with a 0.22 μm filter, the samples were subjected to an Agilent 1260 chromatography system (Agilent 1260, Santa Clara, CA, USA) equipped with a ZORBAX Eclipse XDB-C18 column (Agilent, 4.6 mm × 150 mm, 3.5 µm) and an ultraviolet detector (Agilent 1260, Santa Clara, CA, USA) to detect the produced FA. The system was eluted with a mobile phase composed of acetic acid, methanol, and water with a flow rate of 1 mL/min at 30 °C, as previously described [19]. The standard FA was used for qualitative and quantitative analysis. One unit (U) of FAE activity was defined as the required enzyme amount for releasing 1 μmol FA in 1 min.

For XYN activity measurement, 100 μL of purified XYN was mixed with 100 μL of 10 mg/mL wheat xylan in 0.05 M citric-Na_2_HPO_4_ (pH 6.0) to react at 40 °C for 5 min. The reaction was terminated by adding 200 μL of 3,5-dinitro-2-hydroxybenzoic acid (DNS) [20], followed by boiling for 5 min. The generated reducing sugar was detected at 540 nm. Activities were calculated using xylose as the standard. One unit (U) of XYN activity was defined as the amount of enzyme catalyzing the release of 1 μmol of reducing sugars in 1 min.

### 2.4. Optimizing the Secretion of FAE and XYN

In order to optimize the secretory expression of FAE and XYN, five levels of initial cell density (OD_600_ = 0.5, 0.8, 1.0, 1.5, and 2.0, respectively), five levels of IPTG concentration (0.05, 0.1, 0.2, 0.5, and 1.0 mM, respectively), five levels of induction temperature (16, 20, 25, 30, and 37 °C, respectively), as well as three levels of glycine concentration (2, 5, and 10 g/L, respectively) were all evaluated while maintaining the other parameters at a constant level. Experiments were performed in a 250 mL flask containing 50 mL of the medium. After induction for 12 h, the cell-free supernatants were collected for FAE and XYN activity measurement.

### 2.5. Release of FA by Recombinant E. coli Strains

To assess the FA-release capability of the constructed recombinant *E. coli* strains, the de-starched wheat bran (DSWB) was prepared as previously reported [21], and then added into the LB medium as the substrate. SP1-F was induced with the optimal secretion parameters for FAE, i.e., supplementing 0.2 mM of IPTG and 10 g/L of glycine when the OD_600_ reached 2.0, and then incubating at 30 °C for 48 h. SP4-X was induced with the optimal secretion parameters for XYN, i.e., supplementing 0.5 mM of IPTG and 10 g/L of glycine when the OD_600_ reached 1.0, and then culturing at 30 °C for 48 h. SP1-F-SP4-X was cultured in two batches: one induced with the optimal parameters for FAE (SP1-F-SP4-X1) and the other with those for XYN (SP1-F-SP4-X2), respectively. The SP1-F (OD_600_ = 0.2) and SP4-X (OD_600_ = 1.0) were mixed (SP1-F+SP4-X) and induced with 0.2 mM of IPTG plus 10 g/L of glycine at 30 °C for 48 h. The total amount of FA in the DSWB was determined through alkaline hydrolysis (2.0 M NaOH containing 1% Na_2_SO_3_ for 10 h at 25 °C in dark condition). After acidification with HCl, the samples were extracted using ethyl acetate and then analyzed with HPLC, as described above [22].

### 2.6. Statistical Analysis

The results were presented as the mean ± standard deviation and all experiments were performed in triplicate. The date was analyzed using ANOVA analysis using SPSS version 13.0.

## 3. Results and Discussion

### 3.1. Selection of the Best Signal Peptide for FAE and XYN

It is generally considered that the signal peptide selection is a simple and effective strategy for recombinant protein secretion, aiding to promote its soluble expression, reduce inclusion body formation, improve enzyme activity, and facilitate downstream processing [23,24,25]. In order to realize the secretory expressions of FAE, the *fae* gene was ligated into the pET-22b(+) vector to be fused with the intrinsic signal peptide pelB. Thereafter, the pelB SP was substituted with six different SPs using fusion PCR, including one *Bacillus* alkaline cellulase SP, four *Lactobacillus* FAE SPs, and the osmotically inducible protein Y (OsmY), respectively. After 12 h of induction, these signal peptides were able to assist FAE secretion in most recombinant strains, which were then verified by monitoring the FAE activity in each subcellular location and conducting SDS-PAGE analysis of these extracellular proteins. The SP1-F exhibited the highest activity (0.087 U/mL) among all the strains, and the extracellular activity of other recombinant strains were less than half of this. Subsequently, the more cytoplasmic and periplasmic FAE could be detected and its highest level was detected as 1.17 U/mL. In contrast, the OsmY-F and pelB-F exhibited a relatively low enzymatic activity in all three components, suggesting that OsmY and pelB may cause FAE misfolding, and are not suitable for its expression and secretion (Figure 1A).

The SP2-F, SP3-F, and SP4-F also exhibited high expression levels of FAE; however, most of the enzyme accumulated in the cytoplasm or periplasm, and only a small amount of the enzyme was secreted out of the cell. SDS-PAGE analysis also demonstrated that FAE existed in the cell-free culture supernatant and among which SP1-F had the highest band density (Figure 1B), suggesting that the signal SP1 was the most optimal signal peptide for the extracellular expression of FAE.

The optimal SP for XYN was also selected using the same strategy. All the fused XYN enzymes were able to be expressed and secreted effectively, exhibiting the extracellular XYN enzyme activity ratio from 13.9% to 24.0%, respectively. The SP4-X demonstrated the highest extracellular XYN activity after inducing for 12 h (0.35 U/mL), which was in consistence with its greatest abundance detected during SDS-PAGE analysis. Since the SP4-X showed a high XYN expression level, and the proportion of extracellular enzyme activity was up to 23.6%, SP4 was subsequently selected as the best signal peptide for XYN secretory expression (Figure 2).

The *fae* and *xyn* genes, as well as their best signal peptides SP1 and SP4, were then integrated into the pETDuet-1 vector, generating the expression plasmid pETDuet-SP1-fae-SP4-xyn, which was then verified with PCR using the specific primer pairs F1-*Nde* I/R-*Xho* I-xyn. The strain SP1-F-SP4-X, harboring this plasmid, was consequently constructed for the simultaneous expression of FAE and XYN with the synergic degradation of biomass.

### 3.2. Optimization of the Secretion of FAE and XYN

A large number of factors are known to affect the expression of recombinant proteins in *E. coli*. Although FAE and XYN were able to be expressed and secreted by SP1-F-SP4-X, their activities were too low to cause the release of all the FA from the agricultural residues. Hence, a single-factor design of experiments, including initial cell density (OD_600_), concentration of IPTG, induction temperature, and concentration of glycine in 250mL flasks was applied to further improve their expression. As shown in Figure 3A,B, the initial cell density and IPTG concentration did not show any significant effect on enzyme secretion.

For IPTG induced fast synthesis of recombinant enzymes, the lower temperature could prevent the misfolding and consequently improve the solubility of the enzyme. The optimized induction temperature was 30 °C for FAE and 37 °C for XYN (Figure 3C), respectively. It should be noted that the optimal induction temperature was quite different from the enzyme’s thermal tolerance. The former one mostly decided the performance of the host that expressed, folded, and routed the enzyme; while the latter one played a relatively little role on the stability of the enzyme itself. That is why the FAE enzyme obtained from the cow rumen has been proven to have a great thermal tolerance, but its expression under higher temperatures, such as 37 °C, decreased sharply compared to that observed under 30 °C.

Glycine is known to interfere with peptidoglycan synthesis and therefore affect the integrity and permeability of the outer membrane permeability of *E. coli* [26]. In the present study, it was revealed that for FAE and XYN secretion, glycine played much more important roles than the factors tested above. With the optimal concentration of supplemented glycine (10 g/L), the secretion of FAE and XYN were increased by 5.6 times and 34%, respectively (Figure 3D).

The secretion of FAE was finally optimized to be induced by 0.2 mM of IPTG at 30 °C when the OD_600_ reached 2.0 and supplemented with 10 g/L of glycine. For XYN, the parameters were optimized with the initial OD_600_ of 1.0, 0.5 mM IPTG, 37 °C, and 10 g/L of glycine. Under their optimal conditions, the extracellular activity of FAE and XYN reached 0.94 U/mL and 0.87 U/mL, which were 10.9 and 2.5 times higher than that before their optimization, respectively. These findings could indicate that a large amount of FAE was secreted into the medium, with the extracellular activity accounting for 30.6% of the total FAE activity after optimization, whereas it was only 4.1% before optimization (Figure 4A), respectively. For XYN, its extracellular activity accounted for 51.6% of the total after optimization, which was only 24.5% before optimization, respectively (Figure 4B).

Although FAE and XYN have different optimal culture conditions, their extracellular activities were higher under both of these conditions than that before optimization.

### 3.3. Production of FA from the DSWB Using Recombinant E. coli Strains

The direct use of the recombinant *E. coli* strain would make the degradation of biomass more economically feasible, due to the savings made for the enzyme purification process. The DSWB (4%), containing 3.49 mg/g of FA, was supplemented into the LB medium as a carbon source for *E. coli* cell growth, and also acted as the substrate for producing FA. SP4-X expression only supported the functions of recombinant XYN, and the control BL21 strain harboring the blank pETDuet1 vector was not able to produce any FA, indicating that FAE was essential for releasing FA from the biomass. With FAE alone, 1.64 mg/g DSWB of FA was obtained with the SP1-F strain in 48 h, with the release rate of 47.0%. By the joint actions of FAE and XYN using the strain SP1-F-SP4-X under the FAE optimal secretion condition for 48 h, a 91.1% (3.18 mg/g DSWB) release rate of FA was achieved, producing 127.2 mg/L of FA.

It was noted that a relatively lower yield (release rate) and titer was obtained when the strain was induced under the XYN optimal secretion condition, suggesting that the FAE-catalyzed reaction was the limiting step during FA release. The yield of FA using the mixed strains of SP1-F and SP4-X was almost the same as that with the single strain SP1-F-SP4-X1 (Table 2).

Furthermore, 8% and 10% of DSWB was supplemented in the releasing system, which increased the FA titer to 256.8 mg/L and 314.1 mg/L at 48 h, respectively, without causing a decrease in the yield of FA. In previous studies, actions of the FAE enzyme from *Lactobacillus* alone achieved the highest FA release rate of 1.48 mg/g of DSWB [18,19], and joint actions by the multi-modular, bifunctional XYN/FAE achieved a yield of 2.37 mg/g of DSWB [23]; while in the present study, the release rate was 3.18 mg/g of DSWB, which is the highest to the best of our knowledge.

FA has also been reported to be the precursor of vanillin. With the metabolic pathway from FA to vanillin integrated into the SP1-F-SP4-X1 strain, the tentative plan of vanillin production from agriculture residues, as exemplified by wheat bran, could be established. At the same time, the final productivity of the reducing sugar reached 4023.4 mg/L, indicating that SP1-F-SP4-X also demonstrated a great application potential in the production of reducing sugars (xyloses/xylo-oligosaccharides) [20].

## 4. Conclusions

FA has been widely used as a functional ingredient across the cosmetics, medicine, and food industries. As a natural source of FA and an agricultural residue with a low price, wheat bran has become an excellent raw material for natural ferulic acid production. The present study attempted to recombinantly express the ferulic acid esterase and xylanase with a greater activity and superior tolerance from the metagenomes of the cow and camel rumen, which are treasure troves of glycoside hydrolases, to enzymatically produce FA from agricultural residues. By screening seven candidate signal peptides, the optimal ones, SP1 and SP4, were selected and fused with ferulic acid esterase and xylanase, respectively. After optimizing the inducing conditions of the constructed strain SP1-F-SP4-X, these two enzymes were expressed and secreted successfully, with extracellular activities of 0.84 U/mL and 0.62 U/mL, respectively. When cultured with 10% DSWB, a titer of 314.1 mg/L FA with a release rate of 92.0% was obtained after 48 h of induction. A reasonable quantity of reducing sugar consisting of xylose and xylo-oligosaccharide were also obtained at the same time, indicating the great potential of this constructed strain.

## Figures and Tables

**Figure 1 microorganisms-11-01869-f001:**
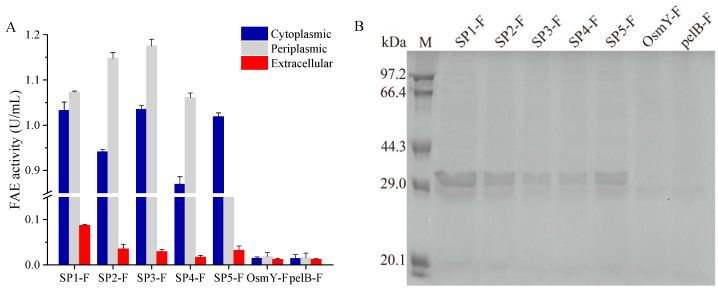
Comparison of the activity of FAE fused with different SPs (**A**), and extracellular protein profiles of recombinant *E. coli* strains carrying different FAE expression vectors (**B**). Data reflect the mean ± SD (*n* = 3).

**Figure 2 microorganisms-11-01869-f002:**
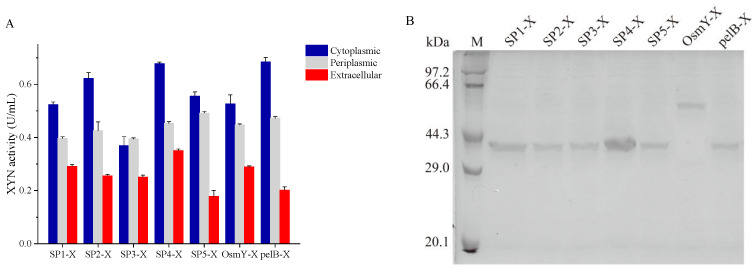
Comparison of the activity of XYN fused with different SPs (**A**), and extracellular protein profiles of recombinant *E. coli* strains carrying different XYN expression vectors. (**B**) Data reflect the mean ± SD (*n* = 3).

**Figure 3 microorganisms-11-01869-f003:**
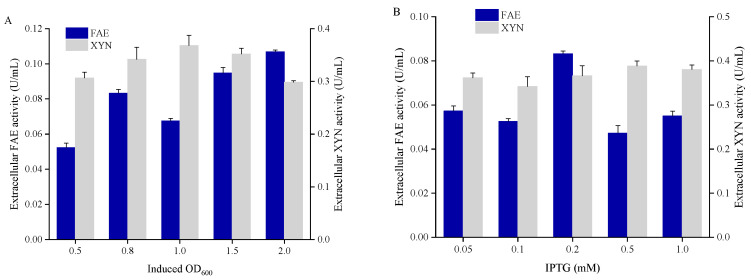
Single-factor analysis on FAE and XYN secretion with (**A**) five levels of initial cell density (OD_600_ = 0.5, 0.8, 1.0, 1.5, and 2.0, respectively), (**B**) five levels of IPTG concentration (0.05, 0.1, 0.2, 0.5, and 1.0 mM, respectively), (**C**) five levels of induction temperature (16, 20, 25, 30, and 37 °C, respectively), and (**D**) three levels of glycine concentration (2, 5, and 10 g/L, respectively). Data reflect the mean ± SD (*n* = 3).

**Figure 4 microorganisms-11-01869-f004:**
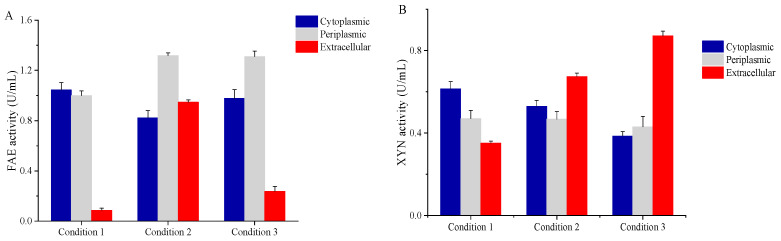
Comparison of FAE (**A**) and XYN (**B**) activity before optimization (condition 1) and under the optimal secretion conditions of FAE (condition 2) and XYN (condition 3). Data reflect the mean ± SD (*n* = 3).

**Table 1 microorganisms-11-01869-t001:** The signal peptides, plasmids, and strains used in this study.

SPs, Plasmids, and Strains	Features	Source
**SPs**		
SP1	MEGNTREDNFKHLLGNDNVK	*Bacillus* sp. [17]
SP2	MSRITIERDGLTLVGDREEP	*Lactobacillus amylovorus* [18]
SP3	MSRVTIERDGLTLVGDREEP	*L. crispatus* [19]
SP4	MSRITIERDSLTLVGDREEP	*L. helveticus* [18]
SP5	MEITIKRDGLKLYGLLEGT	*L. reuteri* [18]
pelB	MKYLLPTAAAGLLLLAAQPAMA	pET22b(+)
OsmY	Genebank: CP060121.1	*E. coli* BL21 [20]
**Plasmids**		
pET22b(+)	Amp^r^	This lab
pETDuet-1	Kan^r^	This lab
pET22b-pelB-*fae*	Amp^r^, pET22b(+) vector ligated with *fae* gene	This study
pETDuet-SP1-*fae*	Amp^r^, pETDuet-1 vector ligated with SP1 and *fae* gene	This study
pETDuet-SP2-*fae*	Amp^r^, pETDuet-1 vector ligated with SP2 and *fae* gene	This study
pETDuet-SP3-*fae*	Amp^r^, pETDuet-1 vector ligated with SP3 and *fae* gene	This study
pETDuet-SP4-*fae*	Amp^r^, pETDuet-1 vector ligated with SP4 and *fae* gene	This study
pETDuet-SP5-*fae*	Amp^r^, pETDuet-1 vector ligated with SP5 and *fae* gene	This study
pETDuet-*osmY*-*fae*	Amp^r^, pETDuet-1 vector ligated with *osmY* and *fae* gene	This study
pET22b-pelB-*xyn*	Amp^r^, pET22b(+) vector ligated with *xyn* gene	This study
pETDuet-SP1-*xyn*	Amp^r^, pETDuet-1 vector ligated with SP1 and *xyn* gene	This study
pETDuet-SP2-*xyn*	Amp^r^, pETDuet-1 vector ligated with SP2 and *xyn* gene	This study
pETDuet-SP3-*xyn*	Amp^r^, pETDuet-1 vector ligated with SP3 and *xyn* gene	This study
pETDuet-SP4-*xyn*	Amp^r^, pETDuet-1 vector ligated with SP4 and *xyn* gene	This study
pETDuet-SP5-*xyn*	Amp^r^, pETDuet-1 vector ligated with SP5 and *xyn* gene	This study
pETDuet-*osmY*-*xyn*	Amp^r^, pETDuet-1 vector ligated with *osmY* and *xyn* gene	This study
pETDuet-SP1-*fae*-SP4-*xyn*	Amp^r^, pETDuet-1 vector ligated with SP1, *fae* gene, SP4 and *xyn* gene	This study
**Strains**		
JM109	*E. coli*, gene cloning	This lab
BL21(DE3)	*E. coli*, gene expression	This lab
*E. coli* pelB-F	BL21 containing pET22b-pelB-*fae*	This study
*E. coli* SP1-F	BL21 containing pETDuet-SP1-*fae*	This study
*E. coli* SP2-F	BL21 containing pETDuet-SP2-*fae*	This study
*E. coli* SP3-F	BL21 containing pETDuet-SP3-*fae*	This study
*E. coli* SP4-F	BL21 containing pETDuet-SP4-*fae*	This study
*E. coli* SP5-F	BL21 containing pETDuet-SP5-*fae*	This study
*E. coli* OsmY-F	BL21 containing pETDuet-*osmY*-*fae*	This study
*E. coli* pelB-X	BL21 containing pET22b-pelB-*xyn*	This study
*E. coli* SP1-X	BL21 containing pETDuet-SP1-*xyn*	This study
*E. coli* SP2-X	BL21 containing pETDuet-SP2-*xyn*	This study
*E. coli* SP3-X	BL21 containing pETDuet-SP3-*xyn*	This study
*E. coli* SP4-X	BL21 containing pETDuet-SP4-*xyn*	This study
*E. coli* SP5-X	BL21 containing pETDuet-SP5-*xyn*	This study
*E. coli* OsmY-X	BL21 containing pETDuet-*osmY*-*xyn*	This study
*E. coli* SP1-F-SP4-X	BL21 containing pETDuet-SP1-*fae*-SP4-*xyn*	This study

**Table 2 microorganisms-11-01869-t002:** Release of ferulic acid and reducing sugars from the DSWB using the recombinant strains.

Strains DSWB Content (%)	FA Release Rate (%)		FA Titer (mg/L)		Reducing Sugar (mg/L)	
	24 h	48 h	24 h	48 h	24 h	48 h
CK, 4%	0	0	0	0	0	0
SP1-F, 4%	36.8 ± 0.4	47.0 ± 0.5	51.40 ± 0.92	65.62 ± 0.64	0	0
SP4-X, 4%	0	0	0	0	1156.01 ± 2.83	1531.32 ± 1.05
SP1-F-SP4-X1, 4%	77.5 ± 0.2	91.1 ± 0.7	108.22 ± 1.36	127.20 ± 1.43	1396.25 ± 1.54	1681.43 ± 1.76
SP1-F-SP4-X2, 4%	35.3 ± 0.6	68.8 ± 0.4	49.20 ± 0.27	96.00 ± 1.39	1636.48 ± 1.08	1966.72 ± 2.16
SP1-F-SP4-X, 8%	70.1 ± 0.3	92.0 ± 0.8	195.70 ± 1.43	256.81 ± 1.52	2627.24 ± 2.37	3573.15 ± 2.71
SP1-F-SP4-X, 10%	65.3 ± 0.1	90.0 ± 0.5	227.82 ± 2.58	314.10 ± 2.48	3137.70 ± 2.55	4023.42 ± 3.43

## Data Availability

No new data were created or analyzed in this study. Data sharing is not applicable to this article.

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
