# Peer review of "Extracellular Expression of Feruloyl Esterase and Xylanase in *Escherichia coli* for Ferulic Acid Production from Agricultural Residues"

_microorganisms, 2023, doi:10.3390/microorganisms11081869_

Round 1

Reviewer 1 Report

This article describes the extracellular expression of feruloyl esterase and xylanase in Escherichia coli and its application for ferulic acid production using agricultural residues as substrates. The authors determined the optimal signal peptides for enzyme secretion and evaluated the efficacy of ferulic acid production from different substrates. The results showed that ferulic acid production was most efficient from de-starched wheat bran. This study may have important implications for sustainable production of ferulic acid from agricultural residues. The work was prepared in a very careful way, it is obvious that you have mastered the scientific technique. Nonetheless, the article must be improved in terms of writing since some grammar and syntax errors are present in the manuscript.

Minor suggestions:

The authors need to revise the title of the paper in a more meaningful way. the title is long and has some unnecessary information, suggestion: "Extracellular Expression of Feruloyl Esterase and Xylanase in E. coli for Ferulic Acid Production from Agricultural Residues";

Please improve the abstract to cover the important topics reviewed and discussed in this article. The abstract is written in a way lacks logic. It should highlight the salient findings more critically;

Keywords are present in the title (ferulic acid; feruloyl esterase; xylanase; Escherichia coli), choose other indexing terms for the article;

The introduction is too long, as well as the paragraphs are huge, this needs to be better.

The report on M&M is very succinct! Provide experimental work plan at the start of M&M. No detail description is available about the experimental design. What statistical analyzes are used?

In figures 1, 3, and table 2, I suggest replacing the standard error with the standard deviation;

The results have long paragraphs. I suggest reducing the size of the paragraphs. The results of this study are not fully explained therefore the interpretation of the results is very difficult. The author needs to provide the % increase or decrease rather than just writing ''significantly increased….''; Authors should discuss the results integrally. The discussion is based on individual results. I suggest that integrating the results will give more value to the work. I suggest that you discuss by integrating all your results. You can use correlation tests (PCA or Pearson Correlation).

The conclusion is totally confusing. Re-write the conclusion! It needs to be much improved.

The work was prepared in a very careful way, it is obvious that you have mastered the scientific technique. Nonetheless, the article must be improved in terms of writing since some grammar and syntax errors are present in the manuscript.

Author Response

1. The authors need to revise the title of the paper in a more meaningful way. the title is long and has some unnecessary information, suggestion: "Extracellular Expression of Feruloyl Esterase and Xylanase in  colifor Ferulic Acid Production from Agricultural Residues";

Response: 

The title has been revised accordingly.

2. Please improve the abstract to cover the important topics reviewed and discussed in this article. The abstract is written in a way lacks logic. It should highlight the salient findings more critically;

Response:

The abstract has been re-written in a more logic way (Lines 10 to 21).

3. Keywords are present in the title (ferulic acid; feruloyl esterase; xylanase; Escherichia coli), choose other indexing terms for the article;

Response: 

Two new keywords ‘signal peptides’ and ‘release rate’ were chosen to replace ‘xylanase’ and ‘E. coli BL21’ (Line 22).

4. The introduction is too long, as well as the paragraphs are huge, this needs to be better.

Response: 

The introduction section has been re-organize to be more concise and readable (Lines 24 to 62).

5. The report on M&M is very succinct! Provide experimental work plan at the start of M&M. No detail description is available about the experimental design. What statistical analyzes are used?

Response: 

The M&M section has been improved. The work plan and experimental design have been added at the start of each part in the M&M section accordingly. The ‘2.6. statistical analysis’ was added in the revised manuscript to explain how the data was analyzed (Lines 139 to 142).

6. In figures 1, 3, and table 2, I suggest replacing the standard error with the standard deviation;

 Response: 

They have been replaced and described in section 2.6.

7. The results have long paragraphs. I suggest reducing the size of the paragraphs. The results of this study are not fully explained therefore the interpretation of the results is very difficult. The author needs to provide the % increase or decrease rather than just writing ''significantly increased….''; Authors should discuss the results integrally. The discussion is based on individual results. I suggest that integrating the results will give more value to the work. I suggest that you discuss by integrating all your results. You can use correlation tests (PCA or Pearson Correlation).

Response: 

The paragraphs size have been reduced. The increases in protein expression, FA release rate and productivity have been described quantitatively in the revised manuscript. Since the results mainly focused on three modules: genetic modulation for extracellular expression of enzymes, optimization of enzyme expression conditions and production of FA using the recombinant E. coli strains, we discussed the results separately in the end of each section, instead of integrating all the results. There seems no need to take a correlation test.

8. The conclusion is totally confusing. Re-write the conclusion! It needs to be much improved.

Response: 

It has been re-written for easier understanding.

Reviewer 2 Report

General comments:

1.       A comparison with other extracellular producers of these enzymes needs to be given in the paper. Namely, according to information in the literature (e.g. https://doi.org/10.1016/j.biortech.2019.121526 for feruloyl esterase and https://doi.org/10.1016/j.procbio.2018.05.019 for xylanse), the production of these enzymes from the selected sources is not very promising.

2.       Moreover, the authors claim that feruloyl esterase is a thermostable enzyme. From the results shown in Fig. 3C, it can be seen that the activity decreases significantly with increasing temperature from 30 to 37°C.

Specific comments:

1.       Figure 3 should be cited in the chapter 3.2.

2.       Figure 3 – In such analyses, it is necessary to specify the exact conditions under which a particular factor is optimised. Therefore, the title of the figure should be upgraded.

3.       It is not clear whether SP1-F-SP4-X is one strain or two strains.

Author Response

Comments on the Quality of English Language

The work was prepared in a very careful way, it is obvious that you have mastered the scientific technique. Nonetheless, the article must be improved in terms of writing since some grammar and syntax errors are present in the manuscript.

Response:

The manuscript has been carefully revised and proofread by all the authors.

General comments:

1. A comparison with other extracellular producers of these enzymes needs to be given in the paper. Namely, according to information in the literature (e.g. https://doi.org/10.1016/j.biortech.2019.121526 for feruloyl esterase and https://doi.org/10.1016/j.procbio.2018.05.019 for xylanse), the production of these enzymes from the selected sources is not very promising.

Response:

The results in this present study haven been compared with previous related researches mentioned above (Lines 253 to 257).

2. Moreover, the authors claim that feruloyl esterase is a thermostable enzyme. From the results shown in Fig. 3C, it can be seen that the activity decreases significantly with increasing temperature from 30 to 37°C.

Response: 

The optimal induction temperature was quite different from the enzyme thermal tolerance, as the induction temperature mostly decided the performance of the host that express, fold, and route enzyme, but played a very little role on the stability of the enzyme. That is why the FAE from cow rumen has been proved to be with great thermal tolerance, but its expression at higher temperature such as 37 °C decreased sharply comparing to that at 30 °C. This discussion has been added in the revised manuscript (Lines 199 to 205).

Specific comments:

1. Figure 3 should be cited in the chapter 3.2.

Response: 

It has been revised accordingly (Lines 212 to 217).

2. Figure 3 – In such analyses, it is necessary to specify the exact conditions under which a particular factor is optimised. Therefore, the title of the figure should be upgraded.

Response: 

The title of Figure 3 has been upgraded (Lines 214 to 217).

3. It is not clear whether SP1-F-SP4-X is one strain or two strains.

Response: 

The strain SP1-F-SP4-X is one strain that could express extracellular FAE and XYN simultaneously. This has been illustrated in Table 1. To avoid any misunderstanding and make the manuscript more clearly and easier to understand, the general characteristics of the generated recombinant strains were also described in the results section 3.1 (Lines 184 to 186).

Round 2

Reviewer 2 Report

The manuscript is appropriately correct and can be accepted in the present form.

Author Response

Point 1: The manuscript is appropriately correct and can be accepted in the present form.

Response: Thank you. We have proof-read the manuscript and revised several places to make it better. The changes are listed in the submitted file. 
